# Research on Fast Recognition and Localization of an Electric Vehicle Charging Port Based on a Cluster Template Matching Algorithm

**DOI:** 10.3390/s22093599

**Published:** 2022-05-09

**Authors:** Pengkun Quan, Ya’nan Lou, Haoyu Lin, Zhuo Liang, Dongbo Wei, Shichun Di

**Affiliations:** School of Mechatronics Engineering, Harbin Institute of Technology, Harbin 150001, China; quanpengkun@hit.edu.cn (P.Q.); louyn@stu.hit.edu.cn (Y.L.); linhaoyu@hit.edu.cn (H.L.); liangzhuo@hit.edu.cn (Z.L.); weidb@hit.edu.cn (D.W.)

**Keywords:** DC charging port for electric vehicles, unmanned charging, pose estimation, non-cooperative features, cluster template matching algorithm

## Abstract

With the gradual maturity of driverless and automatic parking technologies, electric vehicle charging has been gradually developing in the direction of automation. However, the pose calculation of the charging port (CP) is an important part of realizing automatic charging, and it represents a problem that needs to be solved urgently. To address this problem, this paper proposes a set of efficient and accurate methods for determining the pose of an electric vehicle CP, which mainly includes the search and aiming phases. In the search phase, the feature circle algorithm is used to fit the ellipse information to obtain the pixel coordinates of the feature point. In the aiming phase, contour matching and logarithmic evaluation indicators are used in the cluster template matching algorithm (CTMA) proposed in this paper to obtain the matching position. Based on the image deformation rate and zoom rates, a matching template is established to realize the fast and accurate matching of textureless circular features and complex light fields. The EPnP algorithm is employed to obtain the pose information, and an AUBO-i5 robot is used to complete the charging gun insertion. The results show that the average CP positioning errors (*x*, *y*, *z*, *Rx*, *Ry*, and *Rz*) of the proposed algorithm are 0.65 mm, 0.84 mm, 1.24 mm, 1.11 degrees, 0.95 degrees, and 0.55 degrees. Further, the efficiency of the positioning method is improved by 510.4% and the comprehensive plug-in success rate is 95%. Therefore, the proposed CTMA in this paper can efficiently and accurately identify the CP while meeting the actual plug-in requirements.

## 1. Introduction

With the reduction of petroleum energy amounts, new energy-based vehicles have become more important [1]. Electric vehicles have the advantages of producing no harmful gas emissions, being clean and environmentally friendly, having fast acceleration and low cost, and not consuming petroleum energy. Therefore, they have received significant financial support from the governments of many countries and have become the future development direction of most automobile manufacturers. Due to all of these factors, electric vehicles have been developing rapidly in recent years [2,3,4,5], and automatic parking and driverless technologies have gradually matured [6]. Accordingly, electric vehicle users do not need to reach parking spaces and garages. It should be noted that manual charging is impractical, the fast-charging connector cable is heavy and thick, and the plugging force is large. In addition, manual insertion also has potential safety hazards [7]. Therefore, a method for the automatic charging of new-energy electric vehicles is urgently needed.

The main solution to the problem of electric car charging is that a robot realizes an automatic connection between the charging gun and the electric car charging port (CP). At present, a number of research institutions and companies have proposed their own implementation plans for overcoming the mentioned problem [8,9,10,11,12]. Currently, the main strategy for the automatic charging of electric vehicles includes the recognition of a charging port’s pose and robotic insertion machinery. Thus, the recognition and positioning of a CP denote a prerequisite for a robot to complete the charging, and thus ensuring the accuracy and stability of CP recognition is an important guarantee for successful robot charging. Therefore, the realization of rapid and high-precision identification and positioning of a CP is crucial to the automatic charging technology of electric vehicles.

The CP identification methods can be roughly divided into feature recognition-based identification methods and non-feature recognition-based identification methods. Feature recognition means adding a specific mark to a CP to reduce the difficulty of identification, but the CP needs to be modified. In the identification methods without feature recognition, the original CP features are recognized directly without changing the CP. In terms of feature recognition, in [13], the authors added white labels to the four fixed-point positions of a CP, and based on monocular vision, they used feature matching to achieve preliminary positioning and adjusted the visual error according to the six-dimensional force sensor data to achieve the plug-in. However, they did not provide information on positioning accuracy. In [14], five white circular features around a CP were used, and based on the principle of the farthest distance from the center to the contour, the distance from the center of the ellipse was obtained. In addition, the geometric solution method was used to obtain the pose information on the CP. Under the light intensity of 4 klux, the positioning error of this method was 1.4 mm, the angle error was 1.6°, and the insertion success rate was 98.9%. In [15], Halcon commercial vision software and template matching were used to obtain the location of features, and the least square fitting method was employed to calculate the perspective transformation to obtain information on the CP pose, with a successful insertion rate of 80%. In [16], the authors used the surf feature point matching method to perform an indoor target positioning and insertion experiment. The specific positioning accuracy was not given, and the obtained insertion success rate was 96%. Further, in [17], the positioning error of an indoor CP was tested by Halcon commercial vision software using the template matching algorithm based on binocular positioning. The average position error was 2.5 mm, and the average angle error was 0.8°. In [18], the authors used the Hough circle to obtain the contour features of a CP by the monocular vision method. However, the recognition time was long, and the positioning distance was strict. The position error was ± 1 mm, and the angle error was ± 1°. In [19], the recognition accuracy was tested in different scenarios based on monocular vision. The position coordinates of the characteristic points were obtained by the quadratic curve standardization method. In all test scenarios, the average position error was 0.88 mm, and the average angle error was 0.89°.

According to the above-presented research, the positioning of feature points is a key step for obtaining the CP pose. As a common feature acquisition method, a template matching algorithm has been applied to various areas of production and life. However, light, background, and occlusion can affect the accuracy of template matching. When the object size and rotation angle change, the number of templates increases, thereby increasing the template matching time and affecting the matching efficiency. In recent years, a number of studies have achieved certain contributions to the efficiency and robustness of template matching [20,21,22,23,24,25,26,27]. In [20], a template matching algorithm that combines the Gaussian pyramid transform and a particle swarm optimization algorithm, and which can improve the algorithm matching efficiency, was proposed. In [21], the authors proposed an edge template matching algorithm, which uses gradient contour information, adopts a strategy of a directional guide pixel difference, and can reduce the interference of image noise on the matching result. In [22], the authors developed a block matching algorithm, which is based on voting strategy, to reduce the computational burden and a block matching algorithm according to the characteristics of spatial intensity distribution, which can improve the robustness to speckle noise and occluded targets. In [23], a fast template matching algorithm based on the network control, which uses down-sampling to ensure the original accuracy and to improve the matching efficiency, was proposed. Further, in [24], a robust objective function based on the maximum correlation criterion was defined to match features, and then it was optimized for translation and rotation parameters; the effectiveness and robustness of the proposed method were verified. A non-texture target recognition method, which uses the edge layered template matching method to detect and recognize, and thus improves the detection efficiency, was proposed in [25]. In [26], a detector that can detect straight-line contours was proposed for recognition of non-textured parts. The geometric feature gray-scale inversion invariance algorithm based on straight-line contours was used to improve the accuracy and robustness of template matching.

However, the current research on the automatic charging of electric vehicles has still been in the laboratory phase, and the recognition speed of the algorithm is slow. By adding identification, a vehicle has to be modified, which is not conducive to large-scale application. Therefore, this study uses a method of featureless identification. In the search phase, the round recognition positioning method based on the characteristic root is used for rough positioning, and the EPnP algorithm is used to calculate the CP pose. In the aiming phase, for template matching algorithms, most of the above-mentioned algorithms are for images with texture features and linear contour positioning without texture. However, for untextured circular feature images, it is difficult to improve the efficiency and accuracy of the matching algorithm due to the lack of locating corners. Based on the CP characteristics, this paper proposes a cluster template matching algorithm (CTMA) which improves the efficiency and robustness of recognition in a complex-light field and provides a guarantee for accurate positioning of the EPnP algorithm, thereby realizing efficient and accurate CP recognition.

## 2. Materials and Methods

### 2.1. CP Structure and Complex Scene Description

This study considers the fast-charging ports of electric vehicles. The national standard number is GBT20234.3-2011. A CP is shown in Figure 1. The relative position coordinates of a CP are also shown in Figure 1. A CP consists of a total of nine cylindrical holes, which correspond to charging terminal communication 1 (S+), charging terminal communication 2 (S−), vehicle connection confirmation 1 (CC1), vehicle connection confirmation 2 (CC2), DC power supply positive (DC+), DC power supply negative (DC−), low voltage auxiliary power positive (A+), low voltage auxiliary power negative (A−) and DC protective ground (PE). Although the CP considered in this study is of a standard size, the actual use of a CP shows different degrees of wear. In addition, factors such as the manufacturer’s differentiation and surface reflection can interfere with the identification and positioning of a CP. A CP’s surface will also be differentiated under different environments and lighting, which introduces severe difficulties to identification and positioning.

### 2.2. Experimental Platform

The data acquisition and experiment plug-in platform mainly include the control module, vision module, and robot plug-in actuator. The CP image data collection platform is shown in Figure 2. According to the plug-in requirements, the actuator used an AUBO-i5 articulated robot, and the repeat positioning accuracy was 0.02 mm. The camera, from Daheng Image Vision Co., Ltd. (Haidian District, Beijing), was model number MER-125-30GM/C-P, and the resolution was 1292 × 964 pixel. The camera lens model was M0814-MP2 from Combada Company, with a lens size of 8 mm. We used the camera calibration method of Zhang [28], and hand–eye calibration was done using Axelrod’s method [29].

### 2.3. Image Data Collection

To verify the reliability of the algorithm, the weather was divided into sunny and overcast weather, and the time was divided into four time periods: time 1: 8:30–11:30, time 2: 11:30–14:30, time 3: 14:30–17:30, and time 4: 18:00–21:00. Since the indoor daytime test environment was relatively stable, it was subdivided into six situations according to actual scenes, as shown in Table 1 and Table 2. Further, to improve the actual positioning accuracy, the positioning process was divided into two phases, the search phase and the aiming phase, as shown in Figure 3.

The function of the search phase was to find the CP target and to perform rough CP positioning. According to the actual application scenario, the ranges of the *x*, *y*, and *z* directions were set to [−150, 150] mm, [−100, 100] mm, and [250, 550] mm, respectively, and the angular ranges in the *Rx*, *Ry*, and *Rz* directions were set to [−15, 15]°, [−15, 15]°, and [−15, 15]°. The function of the aiming phase was to locate near the focal length. According to the actual application scenario, the ranges of the *x*, *y*, and *z* directions were set to [−5, 5] mm, [−5, 5] mm, and [245, 275] mm, respectively. The angular ranges of the *Rx*, *Ry*, and *Rz* directions were set to [−5, 5]°, [−5, 5]°, and [−5,5]°.

### 2.4. Identification and Positioning Methods

#### 2.4.1. Technical Route

As mentioned above, the recognition process was divided into two phases, the search phase and the aiming phase. In the search phase, the complexity of positioning was high, but the requirements for positioning accuracy were low. Therefore, in this phase, a more adaptable recognition and positioning algorithm was selected. In contrast, in the aiming phase, the complexity of positioning was low, but the requirements for positioning accuracy were high, so in this phase, an algorithm with high positioning accuracy was selected. Based on these requirements, the CP identification process in this paper is shown in Figure 4.

The specific steps presented in Figure 4 are described within the description of the phase they belong to, as set out below.

In the search phase, the following operations are conducted:(1)The image data are collected and converted into a gray image, and bilateral filtering is performed on the obtained gray image.(2)The canny algorithm is used to obtain contours, and the smaller contours are eliminated. We will re-screen the outline based on the length breadth ratio of the minimum outer rectangle.(3)The characteristic root method is used to fit the contour to an ellipse, and eliminate irrelevant ellipses according to the discrimination conditions.(4)When the number of qualified feature points is not less than six, the qualified feature information is converted into a pixel position matrix, the corresponding three-dimensional space position information is formed into a space position matrix, and the EPnP algorithm is used to solve the pose so as to obtain the pose information on the CP relative to the camera.

In the aiming phase, the following operations are conducted:(1)Before the recognition, the robotic arm is inserted into the CP in the robot teaching mode, and then, according to the result of the hand–eye calibration, the robot arm is controlled to pull out the CP.(2)Images are collected in front of the CP as a template, and the developed template extraction software is used to make the feature template and gradient feature template of the CP.(3)During recognition, image information is collected at the aiming position. First, the bilateral filtering is performed on the image, and then the image contour information is extracted using the canny operator.(4)To improve the accuracy and efficiency of template matching, this paper proposes a method based on the CTMA. In the proposed method, the area of each feature point is matched according to the contour information, thereby reducing the matching time and improving the robustness of the template matching algorithm in an unstable light field environment.(5)Effective features are selected according to decision-making conditions; the effective feature center point position is converted into a pixel position matrix, and the position corresponding to the effective feature is transferred into a spatial position matrix. The EPnP algorithm is used to obtain the CP pose relative to the camera.

The robot is guided to complete the plug-in according to the positioning result.

#### 2.4.2. Feature Recognition Method of a CP in Search Phase

In the search phase, a CP has a large deflection angle relative to the camera, and this study uses the characteristic root method to fit the ellipse based on the mathematical characteristics of the ellipse. Namely, the characteristic root method is used to fit the ellipse using the principle of coordinate transformation between the ellipse coordinate system and the measurement coordinate system. The ellipse can be expressed in the standard coordinate system as follows:(1)$$X_{e}\left[\begin{array}{cc} \lambda_{1} & 0\\ 0 & \lambda_{2} \end{array}\right]X_{e}^{T}=X_{e}\left[\begin{array}{cc} a^{-2} & 0\\ 0 & b^{-2} \end{array}\right]X_{e}^{T}=1,$$
where $X_{e}={\left(\begin{array}{cc} x_{e} & y_{e} \end{array}\right)}^{T}$ denotes the ellipse coordinate and *a* and *b* are the semi-major and semi-minor axes of the ellipsoid, respectively.

The conversion relationship between the measurement coordinate *X* and the ellipse coordinate $X_{e}$ is given by:(2)$$X_{e}={\left[\begin{array}{cc} X_{0} & Y_{0} \end{array}\right]}^{T}+R\left(\alpha \right)X$$
where $\begin{array}{cc} (X_{0} & Y_{0}) \end{array}$ represents the translation between the ellipse coordinate system and the measurement coordinate system and R(α) is the rotation matrix, which is defined as:(3)$$R\left(\alpha \right)=\left[\begin{array}{cc} \cos\alpha & -\sin\alpha \\ \sin\alpha & \cos\alpha \end{array}\right]$$

The part where the coordinate of the measuring point does not satisfy the elliptic equation denotes a residual, and the residual equation is given by:(4)$$v_{i}=X_{ei}^{T}AX_{i}-1$$

The Taylor series expansion and linearization are used to obtain the following relationship:(5)$$v_{i}=\frac{\partial v_{i}}{\partial X_{0}}dX_{0}+\frac{\partial v_{i}}{\partial A}d\Lambda +\frac{\partial v_{i}}{\partial \alpha }d\alpha -l_{i}$$

The error equations for all the characteristic points are listed and iteratively solved under the principle of least quadratic, and the characteristic parameters of the ellipse are obtained. By using Equation (2), the conversion relationship between the measured coordinates and the ellipse coordinates can be obtained; in addition, by substituting ${\left(0,0\right)}^{T}$ into the left side of Equation (2), the coordinates of the center of the ellipse in the measurement coordinate system are obtained by using the calculated parameters.

Thus, the position of the ellipse of the CP and the center point of the ellipse can be fitted. The result is shown in Figure 5.

In the aiming phase, the deflection angle and deflection position of a CP are relatively small, and the complexity of this phase is low, but the requirements for positioning accuracy and efficiency are high. Based on the above situation, a template matching algorithm is used to locate the feature points in the aiming phase. Since the accuracy of the template matching algorithm is sensitive to light, rotation, and object size, the CTMA is developed to improve the accuracy and adaptability of template matching.

(1) Template design

The camera is kept at the focal length of the CP so that the axis of the camera is parallel to the axis of the CP. The camera’s exposure is adjusted to obtain stable image information. In the actual exposure algorithm, the average measurement value of an image is adjusted to between 110–140. The specific operation flowchart is shown in Figure 6.

The collected CP images are taken as the original template data, and the self-made template-making software is used to extract the template of each feature, as shown in Figure 7.

Figure 7 shows the preprocessed original image template and the gradient map template, which are stored in the matching file directory. The above step completes the template production process.

(2) Recognition process based on CTMA

To improve the efficiency and accuracy of template matching, the CTMA uses the location information on feature points to match the location information on the feature contour, and then matches the search area. Based on the coordinate system in Figure 1, this study defines DC−, DC+, S−, CC2, S+, CC1, A−, PE, and A+ as features 1–9, respectively. The position and radius information of each feature point are obtained as $\left(x_{on},y_{on},r_{on}\right),\;\mathrm{where}\;n=1,\;2,\;\cdots ,\;9$.

In this study, an image was obtained in the aiming phase, and the contour information was obtained after image preprocessing. The three special features obtained from all contours are feature 1, feature 2, and feature 8. The specific implementation process is as follows.

The feature points that meet the primary selection conditions are screened out, and all the contour pixel positions and the circumscribed rectangle contour information are defined as $\left(x_{pn},y_{pn},w_{pn},h_{pn}\right),\;\mathrm{where}\;n=1,\;2,\;\cdots ,\;n$, thereby establishing feature 1. The contour matching function between features 2 and 8 is given by:(6)$$\left\{ \begin{array}{c} D_{nm}=\left[\sqrt{{\left(y_{on}-y_{om}\right)}^{2}+{\left(x_{on}-x_{om}\right)}^{2}}-\left(r_{om}+r_{on}\right)\right]\cdot \frac{c_{pn}w_{pn}+c_{pn}h_{pn}}{2r_{on}}\\ \sqrt{{\left(x_{pn}-x_{pm}\right)}^{2}+{\left(y_{pn}-y_{pm}\right)}^{2}}=c_{length_{nm}}\left[\left(c_{pn}w_{pn}+c_{pn}h_{pn}+c_{pm}w_{pm}+c_{pm}h_{pm}\right)/4+D_{nm}\right]\\ \sqrt{{\left(x_{pn}-x_{pj}\right)}^{2}+{\left(y_{pn}-y_{pj}\right)}^{2}}=c_{length\_nj}(c_{pn}w_{pn}+c_{pn}h_{pn}+c_{pj}w_{pj}+c_{pj}h_{pj})/2\\ log_{a}\left(1-\left|1-c_{length_{nm}}\right|\right)+log_{a}\left(1-\left|1-c_{length_{nj}}\right|\right)=R\left(n\right) \end{array},\right.$$
where $D_{nm}$ denotes the shortest distance between the circumscribed surfaces of features *n* and *m*; $c_{length_{nm}}$ is the deviation coefficient of features *n* and *m*; $c_{length_{nj}}$ is the deviation coefficient of features *n* and *j*; $c_{pn},\;c_{pm},$ and $c_{pj}$ denote the contour adjustment factors of features *n*, *m*, and *j*, respectively; a represents the coefficient for adjusting the matching trend, which is 10 here; and R(n) represents the matching degree under the nth combination.

According to all obtained contour information, Equation (6) is used to perform contour matching. Based on the contour matching degree R(n), $\left(x_{p1},y_{p1},w_{p1},h_{p1}\right),\;\left(x_{p2},y_{p2},w_{p2},h_{p2}\right),$ and $\left(x_{p8},y_{p8},w_{p8},h_{p8}\right)$ are obtained, and thus the matching preselected positions can be determined. The specific implementation equations are as follows:(7)$$\begin{array}{c} x_{pn}=sin[arccot\left(\frac{y_{on}}{x_{on}}\right)-arccot\left(\frac{2\sqrt{{\left(x_{p1}-x_{p2}\right)}^{2}+{\left(y_{p1}-y_{p2}\right)}^{2}}}{x_{p2}-x_{p1}}\right)\cdot s+arccot\left(\frac{2d_{o1}}{x_{o2}-x_{o1}}\right)\cdot s]\cdot \sqrt{x_{on}{}^{2}+y_{on}{}^{2}}\\ y_{pn}=cos[arccot\left(\frac{y_{on}}{x_{on}}\right)-arccot\left(\frac{2\sqrt{{\left(x_{p1}-x_{p2}\right)}^{2}+{\left(y_{p1}-y_{p2}\right)}^{2}}}{x_{p2}-x_{p1}}\right)\cdot s+arccot(\frac{2d_{o1}}{x_{o2}-x_{o1}})\cdot s]\cdot \sqrt{x_{on}{}^{2}+y_{on}{}^{2}} \end{array}$$
where *s* is the direction of the feature point; the value of *s* for features 1, 3, 4, and 6–8 is one and for features 2, 5, and 9 it is −1.

Thus, the position $\left(x_{pn},y_{pn}\right)$ of the characteristic points of the template matching can be obtained. Then, the area information of the feature can be calculated by:(8)$$\left\{ \begin{array}{c} l_{op}=\frac{\sqrt{{\left(x_{p1}-x_{p2}\right)}^{2}+{\left(y_{p1}-y_{p2}\right)}^{2}}+\sqrt{{\left(x_{p1}-x_{p8}\right)}^{2}+{\left(y_{p1}-y_{p8}\right)}^{2}}}{\sqrt{{\left(x_{o1}-x_{o2}\right)}^{2}+{\left(y_{o1}-y_{o2}\right)}^{2}}+\sqrt{{\left(x_{o1}-x_{o8}\right)}^{2}+{\left(y_{o1}-y_{o8}\right)}^{2}}}\\ x_{rn}\in [x_{pn}-c_{f}\cdot r_{on}l_{op},x_{pn}+c_{f}\cdot r_{on}l_{op}]\\ y_{rn}\in [y_{pn}-c_{f}\cdot r_{on}l_{op},y_{pn}+c_{f}\cdot r_{on}l_{op}] \end{array},\right.$$
where $l_{op}$ is the density of pixels; $x_{rn}$ and $y_{rn}$ are the matching ranges in the *x* and *y* directions, respectively; and $c_{f}$ is the matching area coefficient.

Before template matching, a suitable template needs to be obtained, and the template scaling factors in the *x* and *y* directions are established as follows:(9)$$\{\begin{array}{c} c_{mx}=\frac{\sqrt{{\left(x_{p1}-x_{p2}\right)}^{2}+{\left(y_{p1}-y_{p2}\right)}^{2}}}{\sqrt{{\left(x_{op1}-x_{op2}\right)}^{2}+{\left(y_{op1}-y_{op2}\right)}^{2}}}\\ c_{my}=\frac{\sqrt{{\left(x_{p1}+x_{p2}-x_{p8}\right)}^{2}+{\left(y_{p1}+y_{p2}-y_{p8}\right)}^{2}}}{\sqrt{{\left(x_{op1}+x_{op2}-x_{op8}\right)}^{2}+{\left(y_{op1}+y_{op2}-y_{op8}\right)}^{2}}} \end{array},$$
where $x_{op1},\;\;y_{op1},\;\;x_{op2},\;\;y_{op2},\;\;x_{op8}$, and $y_{op8}$ denote the coordinate information on the templates of features 1, 2, and 8 in the original image, respectively, and $c_{mx}$ and $c_{my}$ are the scaling factors of the template in the *x* and *y* directions, respectively.

In this way, the rotation angle and zoom factor of the template are obtained, thereby providing the best conditions for template matching. This study uses the normalized square method for the evaluation of template matching. The matching decision conditions are given by Equation (10):(10)$$R\left(x,y\right)=\frac{\sum_{x^{\prime }y^{\prime }}{\left[T\left(x^{\prime },y^{\prime }\right)\cdot I\left(x+x^{\prime },y+y^{\prime }\right)\right]}^{2}}{\sqrt{\sum_{x^{\prime },y^{\prime }}T{\left(x^{\prime },y^{\prime }\right)}^{2}}\cdot \sqrt{\sum_{x^{\prime },y^{\prime }}I{\left(x+x^{\prime },y+y^{\prime }\right)}^{2}}},$$
where $T\left(x^{\prime },y^{\prime }\right)$ is the pixel value of the template image at $\left(x^{\prime },y^{\prime }\right)$ and $I\left(x+x^{\prime },y+y^{\prime }\right)$ is the original image’s pixel value at $\left(x+x^{\prime },y+y^{\prime }\right)$.

By performing the above-presented operations on the preprocessed and gradient images, the position information on the feature points is obtained. The final matching decision condition is given by:(11)$$\{\begin{array}{c} \begin{array}{cc} x_{apn}=\frac{x_{hpn}+x_{gpn}}{2} & \left|x_{hpn}-x_{gpn}\right| \end{array}<\Delta error1\\ \begin{array}{cc} y_{apn}=\frac{y_{hpn}+y_{gpn}}{2} & \left|y_{hpn}-y_{gpn}\right| \end{array}<\Delta error1\\ \left|x_{pn}-x_{apn}\right|<\Delta error2\\ \left|y_{pn}-y_{apn}\right|<\Delta error2 \end{array},$$
where $\left(x_{hpn},y_{hpn}\right)$ is the matching coordinate value of the feature point in the original image, $\left(x_{gpn},y_{gpn}\right)$ is the matching coordinate value of the feature point in the gradient image, $\left(x_{apn},y_{apn}\right)$ is the final coordinate value of the feature point; $\left(x_{pn},y_{pn}\right)$ is the coordinate value of the contour feature point, Δerror1 is the allowable error range of a single matching degree, and Δerror2 is the allowable error range of the final matching degree.

The specific operation process and results of the CTMA are shown in Figure 8.

#### 2.4.3. CP Pose Calculation

By using the above-mentioned algorithm, the pixel position information of the effective feature points can be obtained, and when combined with the space coordinate position of a CP, it can be transformed into a PNP problem [30]. This study uses the pixel coordinates $\left(x_{apn}\;\mathrm{and}\;y_{apn}\right)$ and the corresponding spatial position coordinates $\left(x_{on},y_{on},\;\mathrm{and}\;z_{on}\right)$ to obtain the position information $\left(x_{pos},y_{pos},\;\mathrm{and}\;z_{pos}\right)$ and angle information $\left(x_{ang},y_{ang},\;\mathrm{and}\;z_{ang}\right)$. To solve the PNP problem, different solving methods require different effective feature points.

Therefore, to improve the positioning accuracy, based on at least six feature points in the space, the space coordinate point vector and the pixel coordinate point vector are established, and the position of a coordinate point in the space can be expressed by setting the weighting factor as follows:(12)$$\{\begin{array}{c} p_{m}^{w}=\sum_{n=1}^{n}\alpha_{mn}c_{n}^{w}\\ \sum_{n=1}^{n}\alpha_{mn}=1 \end{array},$$
where $p_{m}^{w}$ is the known three-dimensional coordinate point in the world coordinate system, $c_{n}^{w}$ is the *n*th feature point of $p_{m}^{w}$ in the world coordinate system, and $\alpha_{mn}$ is the weighting factor.

The positioning process of the EPnP algorithm is as follows:

(1)Select at least four feature points in the world coordinate system;(2)Calculate the weighting factor $\alpha_{mn}$;(3)Calculate the feature points in the camera coordinate system;(4)Calculate the minimum error by the Gauss–Newton algorithm and define the error as follows:
(13)$$error=\underset{{\left(m,n\right)}_{s_{\cdot }t_{\cdot }m}<n}{\sum }\left(\left|\left|c_{m}^{c}-c_{n}^{c}\left|{|}^{2}-\right|\right|c_{m}^{w}-c_{n}^{w}\right|{|}^{2}\right),$$where $c_{m}^{c}$ is the *n*th feature point of $c_{m}^{w}$ in the camera coordinate system;(5)Obtain the three-dimensional coordinates of the feature in the camera coordinate system;(6)Calculate the translation vector ***T*** and rotation matrix ***R*** of the CP pose;(7)The *x*, *y*, and *z* values of the CP pose are the components of the translation vector ***T***; and(8)Solve Equation (14) to obtain the rotation values of the CP pose, namely, *Rx*, *Ry*, and *Rz*:
(14)$$\left[\begin{array}{ccc} \cos Rz & -\sin Rz & 0\\ \sin Rz & \cos Rz & 0\\ 0 & 0 & 1 \end{array}\right]\left[\begin{array}{ccc} \cos Ry & 0 & \sin Ry\\ 0 & 1 & 0\\ -\sin Ry & 0 & \cos Ry \end{array}\right]\left[\begin{array}{ccc} 1 & 0 & 0\\ 0 & \cos Rx & -\sin Rx\\ 0 & \sin Rx & \cos Rx \end{array}\right]=R_{3\times 3}.$$

## 3. Results

### 3.1. CP Pose Error

During data collection, to obtain the actual pose information on a CP relative to the camera, the specific operations were performed:

(1)The world coordinates of the robot base and the CP were kept unchanged.(2)When the robot was in the state of teaching, the charging gun was moved into the CP, and this pose was used as the robot’s zero pose.(3)The charging gun was moved out of the CP, and the robot moved randomly within the recognition range to obtain image information.(4)Based on the zero-pose information and pose information of the robot, the pose information of the camera relative to the CP was obtained, which denoted the actual pose information of the camera relative to the CP.(5)The absolute value of the difference between the actual pose information and the theoretical pose was used as a basis for error judgment.

### 3.2. Search-Phase Pose Accuracy Test

The search phase can be roughly divided into CP feature recognition and pose calculation. The recognition effect results of feature points are presented in Figure 9. The pose error results of the CP are given in Table 3.

According to the results in Table 3, it can be concluded that the errors could be divided into three categories: (1) indoor–night pose errors that were small; the positioning accuracy was the highest and the average pose errors in the directions of (x, y, z,Rx,Ry, and Rz) were 1.57 mm, 1.80 mm, 2.02 mm 1.96°, 2.17°, and 1.67°, respectively; (2) outdoor–overcast condition pose errors; the error at noon was relatively large and the overall pose error was small. The average pose errors in the directions of (x,y, z,Rx,Ry,and Rz) were 1.75 mm, 1.86 mm, 2.16 mm, 2.10°, 2.19°, and 1.87°, respectively; (3) outdoor–sunny day pose errors that were the largest, especially at noon; the average outdoor pose errors in the directions of (x,y,z, Rx,Ry, and Rz) were 2.22 mm, 2.27 mm, 2.58 mm, 2.53°, 2.68°, and 1.97°, respectively. According to the obtained results, the change and intensity of light had a significant impact on the position and orientation of the CP. Under strong light, the CP surface and the light form a certain angle, making the illumination uneven, and thus causing errors in feature positioning or even resulting in unrecognizable situations.

### 3.3. Aiming-Phase Pose Accuracy Test

The aiming phase can be mainly divided into feature recognition and pose calculation. The recognition effect results of feature points in different scenarios are presented in Figure 10. The theoretical pose information was obtained by the pose calculation algorithm, and the actual pose error information was obtained based on the error judgment basis. The pose errors of the CP in different scenarios are shown in Table 4.

Based on the results in Table 4, it can be concluded that the errors could be divided into three categories: (1) indoor–night pose positioning errors that were relatively small, achieving the highest positioning accuracy; the average pose errors in the directions of (x,y, z, Rx,Ry,and Rz) were 0.52 mm, 0.66 mm, 1.05 mm, 1.01°, 0.73°, and 0.42°, respectively; (2) outdoor–overcast condition pose positioning errors; the error at noon was relatively large; the overall pose error was small; the average pose errors in the directions of (x,y, z,Rx,Ry,and Rz) were 0.65 mm, 0.80 mm, 1.24 mm, 1.09°, 0.84°, and 0.51°, respectively; (3) sunny–outdoor condition pose positioning errors; the overall pose error increased significantly, especially at noon; the average outdoor pose errors in the directions of (x,y, z, Rx,Ry, and Rz) were 0.80 mm, 1.08 mm, 1.42 mm, 1.24°, 1.29°, and 0.73°, respectively. Based on the presented results, it can be concluded that during the aiming phase, the external light caused great interference to the positioning. The influence of the external light on the aiming phase was much greater than the influence on the search phase, and it was the dominant affecting factor of pose positioning.

### 3.4. Charging Gun Insertion Test Verification

As the above-presented test results show, the indoor and night test results were basically the same, the outdoor overcast test results were basically the same, and the outdoor sunny test results were basically the same, and thus the test was divided into three cases: (1) indoor case (sunny/overcast time 1/time 2/time 3/time 4); (2) outdoor–sunny case (time 1/time 2/time 3); and (3) outdoor–overcast case (time 1/time 2/time 3). We tested 100 sheets for each condition, as shown in Table 5.

Aiming at the above test plan and using the proposed identification and localization method, the test was conducted using an AUBO-i5 robot as an actuator. The results are shown in Table 6.

Based on the test results, for the indoor case (sunny/overcast time 1/time 2/time 3/time 4), the average plug-in rate of the CP was 99%; for the outdoor–sunny case (time 1/time 2/time 3), the average plug-in rate of the CP was 92%; and lastly, for the outdoor–overcast case (time 1/time 2/time 3), the average plug-in rate of the CP was 94%. The positioning accuracy is positively correlated with the success rate of plug-in.

## 4. Discussion

### 4.1. Results Comparison

To evaluate the performance and advanced nature of the proposed method, the algorithm was compared with the two most advanced CP pose positioning methods. The method was implemented in Python programming language with PyCharm 2017 on a personal computer equipped with an Intel^®^ Core™ i5-6300HQ processor and 16 GB of memory. The running time was the average time of all test data. The comparison results are given in Table 7.

Based on the results in Table 5, the combination of the feature recognition algorithm based on this paper and the EPNP algorithm has the highest positioning accuracy, and the proposed CP identification and positioning method overperformed the comparison methods regarding the identification efficiency. In terms of positioning accuracy, the overall positioning accuracy of the proposed method was improved during the search phase. In the aiming phase, the average positioning accuracy of the proposed algorithm was improved compared with that of Yin’s algorithm [18] and was basically the same as that of Quan’s algorithm [19]. In the two phases, the overall recognition efficiency is 510.4% higher than the current optimal method. The proposed algorithm achieved obviously higher recognition accuracy and efficiency than the other two algorithms.

### 4.2. System Error

The system error can be mainly divided into two parts. The first part is the error generated by a robot, and the repeated positioning accuracy of the robot in this study was about 0.05 mm. The second part is the error of the photographing process—the disturbance of the base and the impact during plug-in can affect the zero-position pose, thus reducing the positioning accuracy.

### 4.3. Feature Point Recognition Deviation

#### 4.3.1. Feature Point Recognition Deviation in Search Phase

In the search phase, the main factors that affect the location of feature points are kernel selection during bilateral filtering and selection of contour height and low thresholds when obtaining contour information. When the inclination angle was too large, the characteristic arc chamfers overlapped; due to the CP contour deformation and wear, the contour fitting deviation occurred.

#### 4.3.2. Feature Point Recognition Deviation in Aiming Phase

In the aiming phase, the proposed CTMA was used for feature point recognition. Based on the exposure algorithm proposed in this paper, the image can be quickly adjusted to the appropriate exposure, and the logarithmic function evaluation standard is used to match the position of contour features, which can reduce the large deviation of template matching position caused by uneven illumination of the image, as shown in Figure 11. It should be noted that conventional template matching methods had large matching errors under both gradient and gray map matchings. The main reason for this was that under the external light, the inclination of the light source and the similarity of features caused the failure of the conventional template matching methods. Compared with the current advanced matching methods, our method improves the matching efficiency and accuracy in the complex light field environment, and the matching robustness is also significantly improved.

The influence of the type and inclination of the light source on the recognition result is shown in Figure 12. In the actual environment, the light source characteristics of a CP are mainly defined by three factors: direct sunlight (A), ambient scattered light (B), and lighting (C). D, E, F and O represent the vertices that make up the sun’s altitude angle $\theta_{1}$ and the sun’s deflection angle $\theta_{2}$. At night, the light mainly comes from lighting (C). In the case of indoors and outdoors under overcast conditions, the light source mainly comes from the scattered light (B) of the environment. When it is sunny outdoors, the light source mainly comes from direct sunlight (A), while the light from lighting (C) is covered, and thus a CP will be affected by the sun’s altitude angle $\theta_{1}$ and the sun’s deflection angle $\theta_{2}$. The angles $\theta_{1}$ and $\theta_{2}$ cause different degrees of shadows in CP features, which is the main reason for the recognition deviation; particularly, in the template matching method, the matching accuracy is relatively significantly lower in this than in other recognition scenarios. This causes a severe problem for template matching algorithms. Although the mentioned factors can affect the matching effect, compared with the results before the improvement, the accuracy of feature matching is significantly improved under the proposed CTMA, which can successfully solve the problem of poor robustness of the template matching algorithms in complex scenarios.

### 4.4. Feature Point Pose Calculation Error

When solving the pose by using the EPnP method, the positioning accuracy mainly depends on the three-dimensional space position of the feature point and the coordinate position of the feature point pixel. During the plug-in process of a CP, different degrees of deformation and wear can be caused, resulting in changes in the three-dimensional space position of the CP. In the process of pose calculation, the spatial mapping relationship changes, which affects the pose calculation accuracy.

### 4.5. Calibration Influence on Result Accuracy

The positioning process involves camera parameter calibration and hand–eye parameter calibration. The camera calibration accuracy is mainly affected by the calibration board, calibration picture definition, and calibration method. This part of the impact is very small. The hand–eye calibration error is mainly the error of the matrix conversion between the end of the robot arm and camera position, and this error is relatively small.

Therefore, the location of feature points is the major factor that affects the pose identification accuracy.

## 5. Conclusions

In this paper, an identification and positioning system of a CP of an electric vehicle is proposed, and the positioning accuracy of the proposed system is tested in different scenarios. The plugging test of the charging gun is completed under the drive of the robotic arm. Compared with the advanced CP positioning methods, in the search phase, the proposed method uses the feature root algorithm to fit the gradient map and the preprocessed image contour to obtain the position information of the feature points. In the aiming phase, based on the proposed CTMA, the logarithmic function discriminant method is used to achieve contour matching, and precise matching is performed in the contour area to obtain the position information on feature points. The CP poses were calculated using the EPnP algorithm, and the manipulator is adjusted to complete the insertion work according to the obtained CP pose.

According to the test results, it can be concluded that the indoor, night pose accuracy is the highest, and the pose accuracy is the worst in the outdoor environment on sunny days. The average positioning accuracy values in the directions of (x, y, z,Rx,Ry, and Rz) are 0.65 mm, 0.84 mm, 1.24 mm, 1.11°, 0.95°, and 0.55°, respectively. The results show that the success rate of insertion in different scenarios is positively correlated with the positioning accuracy. The success rates of indoor and outdoor insertions are 99% and 93%, respectively. Compared with the advanced recognition methods, while improving the recognition accuracy, the proposed method achieves significantly improved recognition efficiency, which provides a guarantee for efficient and accurate CP recognition.

In the future, the proposed recognition algorithm could be optimized to achieve a good recognition effect even on a sunny day outdoors and to further improve the positioning efficiency and robustness of the proposed algorithm.

## Figures and Tables

**Figure 1 sensors-22-03599-f001:**
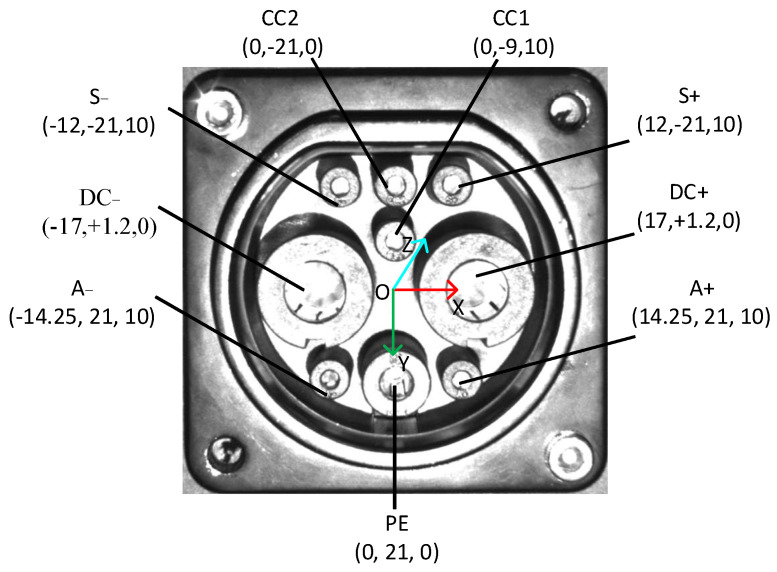
Internal structure diagram and relative position coordinates of a CP.

**Figure 2 sensors-22-03599-f002:**
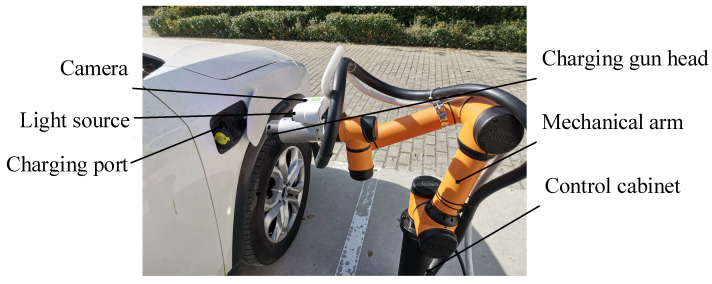
Plug-in experiment platform.

**Figure 3 sensors-22-03599-f003:**
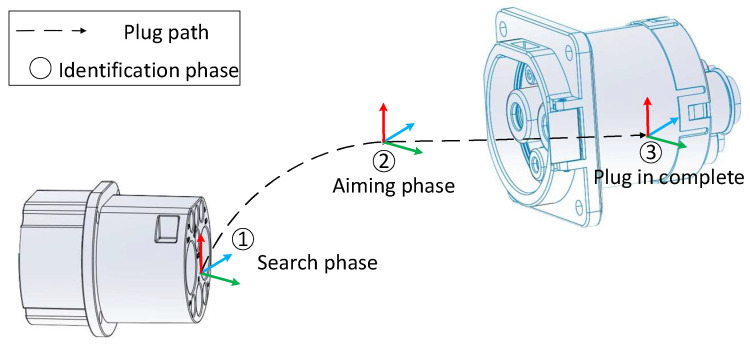
Illustration of identifying the connection path diagram.

**Figure 4 sensors-22-03599-f004:**
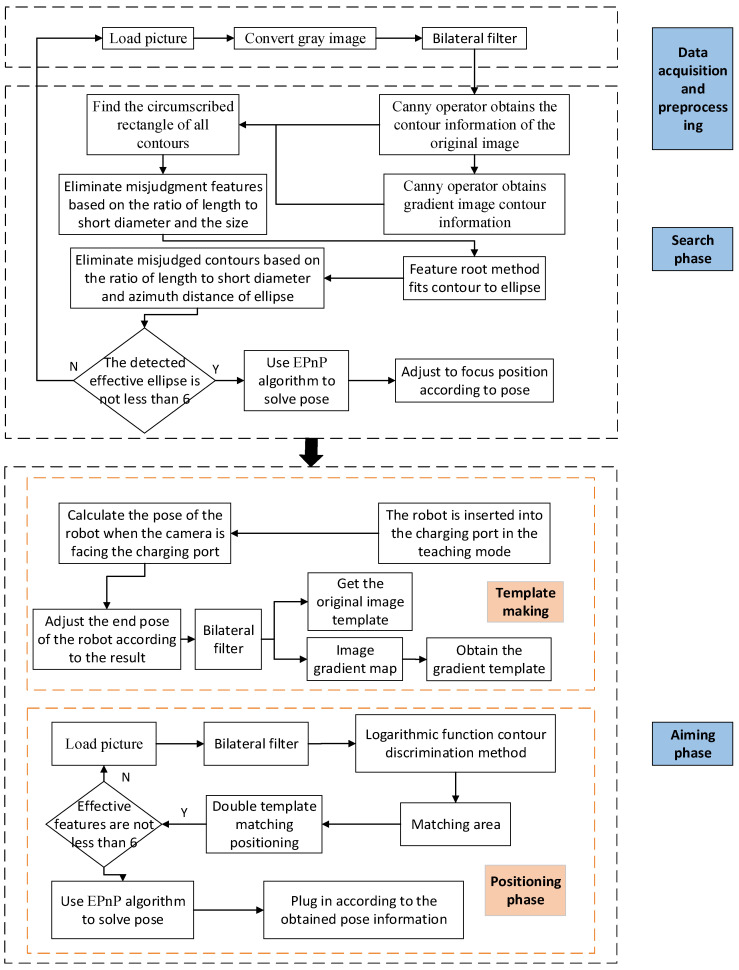
Flow chart of the CP identification method.

**Figure 5 sensors-22-03599-f005:**
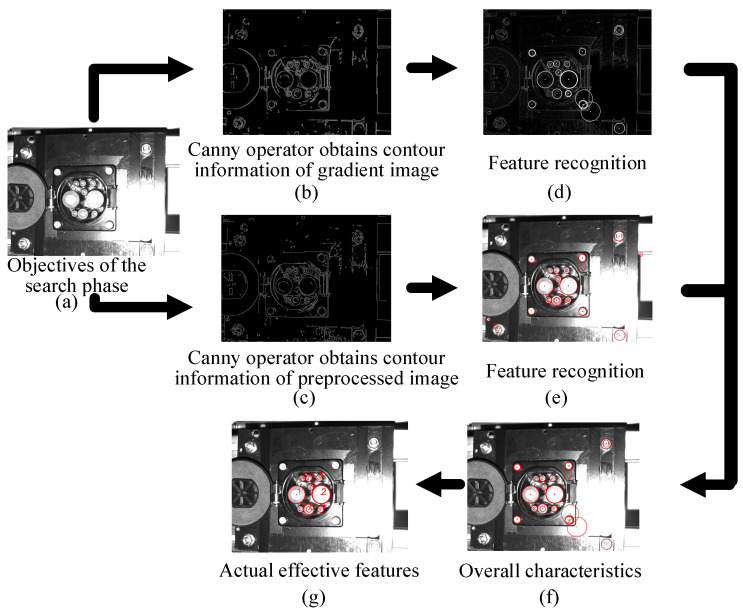
The process of obtaining feature points.

**Figure 6 sensors-22-03599-f006:**
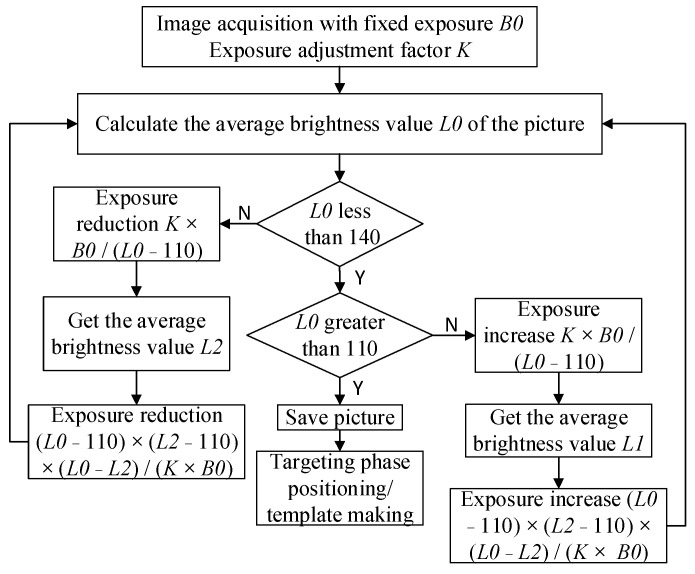
The flowchart of the automatic exposure algorithm.

**Figure 7 sensors-22-03599-f007:**
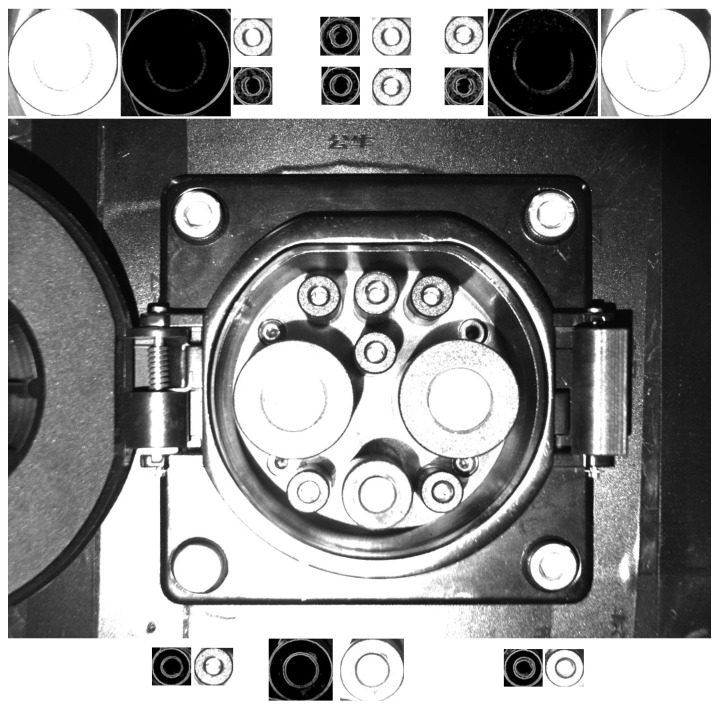
Template extraction example.

**Figure 8 sensors-22-03599-f008:**
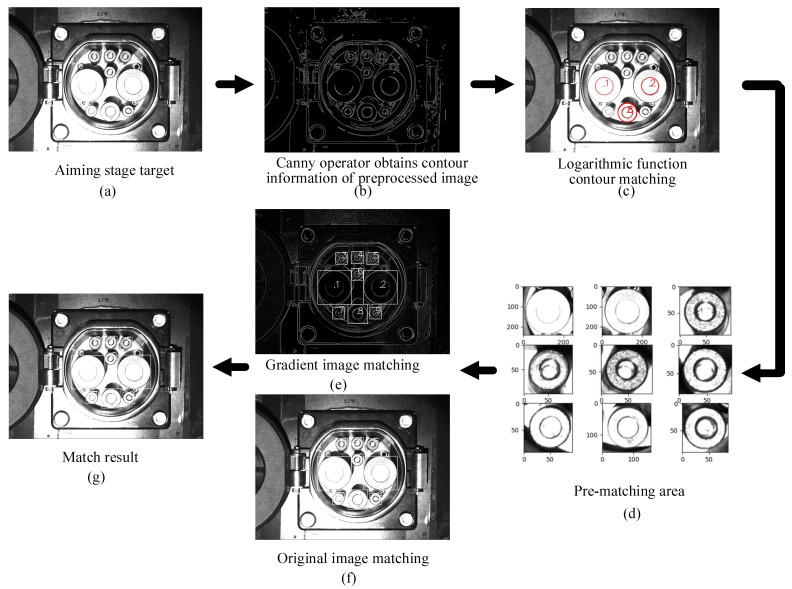
Obtaining feature points.

**Figure 9 sensors-22-03599-f009:**
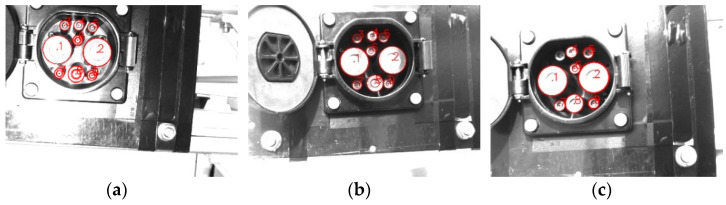
The recognition effect results in different scenarios in the search phase. (**a**) Indoor sunny day/overcast day time 1/time 3. (**b**) Outdoor sunny day time 1/time 3. (**c**) Outdoor sunny day time 2. (**d**) Outdoor overcast day time 1/time 3. (**e**) Outdoor overcast day time 2. (**f**) Time 4.

**Figure 10 sensors-22-03599-f010:**
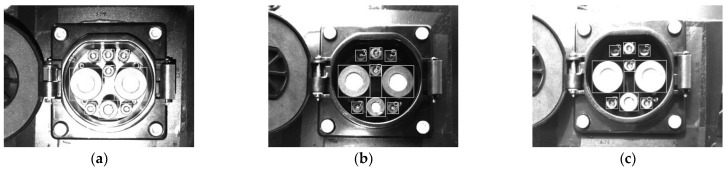
The recognition effect results in different scenarios in the aiming phase. (**a**) Indoor sunny day/overcast day time 1/time 3. (**b**) Outdoor sunny day time 1/time 3. (**c**) Outdoor sunny day time 2. (**d**) Outdoor overcast day time 1/time 3. (**e**) Outdoor overcast day time 2. (**f**) Time 4.

**Figure 11 sensors-22-03599-f011:**
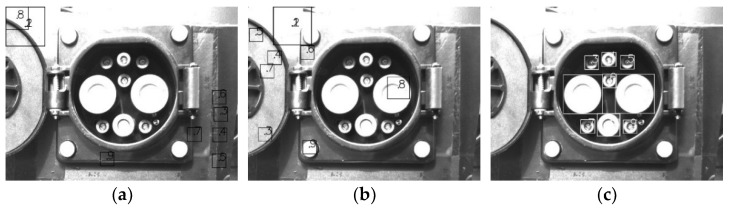
The recognition effect of the template matching method. (**a**) Gradient map template matching before improvement. (**b**) Gray map template matching before improvement. (**c**) Template matching method after improvement.

**Figure 12 sensors-22-03599-f012:**
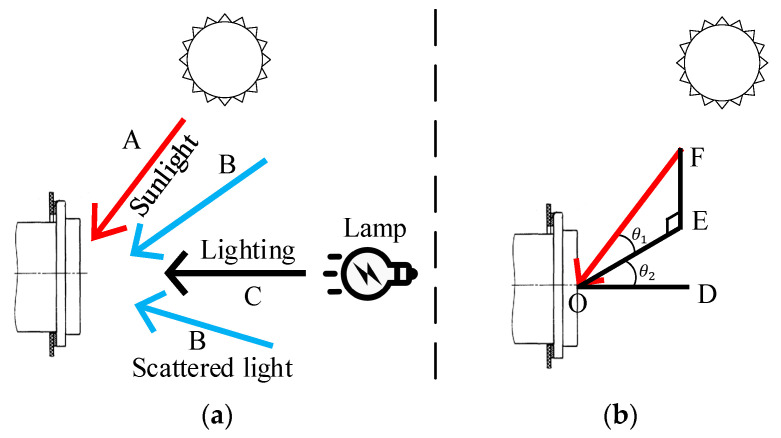
Light field of the CP. (**a**) Overall illumination source of the CP. (**b**) Angle component diagram of sunlight.

**Table 1 sensors-22-03599-t001:** Data information in the search phase.

Scenes	Weather	TimePeriod	Min Light Intensity (Klux)	Max Light Intensity (Klux)	Number of Samples
Indoor	Sunny/Overcast	Time 1/2/3	3.1	4.8	100
Outdoor	Sunny	Time 1/3	7.4	43.8	100
	Sunny	Time 2	12.9	52.3	100
	Overcast	Time 1/3	7.3	15.4	100
	Overcast	Time 2	5.4	22.8	100
Indoor/Outdoor	Sunny/Overcast	Time 4	0.7	3.0	100

**Table 2 sensors-22-03599-t002:** Data information in the aiming phase.

Scenes	Weather	TimePeriod	Min Light Intensity (Klux)	Max Light Intensity (Klux)	Number of Samples
Indoor	Sunny/Overcast	Time 1/2/3	4.5	5.4	100
Outdoor	Sunny	Time 1/3	8.3	43.5	100
	Sunny	Time 2	12.6	52.4	100
	Overcast	Time 1/3	6.9	17.1	100
	Overcast	Time 2	7.2	21.7	100
Indoor/Outdoor	Sunny/Overcast	Time 4	2.9	3.5	100

**Table 3 sensors-22-03599-t003:** Pose solution errors in the search phase.

Scenes	Weather	Time Period	*x*, mm	*y*, mm	*z*, mm	*Rx*, Deg	*Ry*, Deg	*Rz*, Deg
Indoor	Sunny/Overcast	Time 1/2/3	1.62	1.77	1.98	2.10	2.19	1.65
Outdoor	Sunny	Time 1/3	2.12	2.17	2.45	2.44	2.48	1.94
	Sunny	Time 2	2.31	2.36	2.71	2.61	2.87	1.99
	Overcast	Time 1/3	1.73	1.89	2.11	2.07	2.16	1.85
	Overcast	Time 2	1.76	1.82	2.21	2.13	2.21	1.88
Indoor/Outdoor	Sunny/Overcast	Time 4	1.51	1.82	2.05	1.82	2.15	1.69

**Table 4 sensors-22-03599-t004:** Pose errors in the aiming phase.

Scenes	Weather	Time Period	*x*, mm	*y*, mm	*z*, mm	*Rx*, Deg	*Ry*, Deg	*Rz*, Deg
Indoor	Sunny/Overcast	Time 1/2/3	0.52	0.67	1.04	1.07	0.77	0.41
Outdoor	Sunny	Time 1/3	0.74	1.05	1.32	1.21	1.23	0.7
	Sunny	Time 2	0.85	1.11	1.51	1.26	1.34	0.75
	Overcast	Time 1/3	0.62	0.75	1.22	1.04	0.79	0.44
	Overcast	Time 2	0.68	0.84	1.26	1.13	0.89	0.57
Indoor/Outdoor	Sunny/Overcast	Time 4	0.51	0.64	1.06	0.94	0.69	0.43

**Table 5 sensors-22-03599-t005:** Plug-in data information.

Positioning Phase	Scenes	Weather	Time Period	Min Light Intensity (Klux)	Max Light Intensity (Klux)	Number of Samples
Search phase	Indoor	Sunny/Overcast	Time 1/2/3/4	1.74	3.68	100
	Outdoor	Sunny	Time 1/2/3	10.23	48.09	100
		Overcast	Time 1/2/3	6.29	18.90	100
Aiming phase	Indoor	Sunny/Overcast	Time 1/2/3/4	3.92	4.59	100
	Outdoor	Sunny	Time 1/2/3	10.48	48.12	100
		Overcast	Time 1/2/3	6.95	19.36	100

**Table 6 sensors-22-03599-t006:** Results of the plug-in experiment.

Positioning Phase	Scenes	Weather	Time Period	Successfully Identified/Plugged (Times)	Successful Recognition/Plugging Rate (%)
Search/Aiming phase	Indoor	/	/	99	99
	Outdoor	Sunny	AM/PM	92	92
		Overcast	AM/PM	94	94

**Table 7 sensors-22-03599-t007:** Comparison of pose positioning results.

Positioning Phase	Method	*x*, mm	*y*, mm	*z*, mm	*Rx*, Deg	*Ry*, Deg	*Rz*, Deg	Running Time (s)
Search phase	Our + AP3P	2.24	2.46	3.15	6.56	4.21	2.01	0.27
	Our + P3P	2.23	2.47	3.14	5.67	4.11	1.99	0.27
	Our + UPNP	1.88	1.97	2.28	2.34	2.39	1.88	0.27
	Our + ITERATIVE	1.91	1.99	2.34	2.39	2.41	1.91	0.27
	Our + EPNP	1.84	1.97	2.25	2.20	2.34	1.83	0.27
	Quan [18]	2.27	2.53	2.67	/	/	/	1.72
	Yinkai [19]	/	/	/	/	/	/	1.14
Aiming phase	Our + AP3P	1.13	1.32	2.13	4.45	2.34	0.71	0.21
	Our + P3P	1.10	1.34	2.11	4.16	2.15	0.69	0.21
	Our + UPNP	0.66	0.85	1.26	1.13	0.98	0.55	0.21
	Our + ITERATIVE	0.81	0.89	1.33	1.25	1.13	0.61	0.21
	Our + EPNP	0.65	0.84	1.24	1.11	0.95	0.55	0.21
	Quan [18]	0.67	0.88	1.26	1.24	1.01	0.58	1.21
	Yinkai [19]	0.89	1.11	1.31	1.23	1.14	0.63	6.75

## Data Availability

Not applicable.

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
