# Peer review of "Research on Fast Recognition and Localization of an Electric Vehicle Charging Port Based on a Cluster Template Matching Algorithm"

_sensors, 2022, doi:10.3390/s22093599_

Round 1
Reviewer 1 Report
1.131 Fig.1 description - strange description
Table 1,2,3,... AM. PM. <- Please use standard abbreviations https://en.wikipedia.org/wiki/12-hour_clock (Abbreviations section)
Eq.1 - why some parts of this matrix (?) are empty
Eq.2,3 - wrong brackets
l.251 font size failure
Fig.6 why \star operator instead \cdot ?
l.271 ... <- \cdots
Table 1 3 <- 3.0
Eq.7 sign: not clear in this context
Eq.7 and more - brackets problems
Table.1,2 - what is "Weather All" ?
Fig.12 - missing subdescriptions for both images
l.534: "Compared with the advanced recognition methods, while improving the recognition accuracy, the proposed method achieves significantly improved recognition efficiency, which provides a guarantee for efficient and accurate CP recognition"
Comparing should be in Discussion part and related to literature
Author Response
Please see the attachment
We tried our best to improve the manuscript and made some changes in the anuscript.These changes will not influence the content and framework of the paper.
We appreciate for Editors/Reviewers'warm work earnestly,and hope that the orrection will meet with approval.
Once again,thank you very much for your comments and suggestions

Reviewer 2 Report
This paper presents a cluster template matching algorithm for identification and positioning system of a charging port of an electric vehicle. The paper is well-written and interesting. Following concerns may help improving the quality of the paper.
1. Change the caption of Figure 1.
2. Use same font size throughout the paper. Kindly check formatting guidelines of the journal.
3. Please mention reference number against quan and yinkai methods in Table 5.
Author Response

(The authors gave the same response as above.)

Reviewer 3 Report
This paper presents an automatic electric vehicle charging system based on a cluster template matching algorithm (CTMA). The proposed system consists of a search phase and an aiming phase. The goal of the search phase is to obtain the pixel coordinates of the charging port, and the goal of the aiming phase is to match the position based on the proposed CTMA. The proposed system has great potential for practical applications. I have several suggestions.
- Please carefully revise the English writing of this paper before publication. For example, the word in the title “Location” should be “Localization”.
- In Eq. (6), the symbols a and R are not clearly defined. For the term loga(), a is more like the base of the log function, but the authors define it as an adjustment coefficient. Also, R(n) is more like a function, but the authors define it as the degree of contour matching.
- In Eq. (7), the symbol sign: is undefined.
- The proposed method uses a normalized template matching method shown in Eq. (10), which is the same as the TM_CCORR_NORMED template matching method in OpenCV (https://docs.opencv.org/3.4/de/da9/tutorial_template_matching.html). Moreover, the EPnP method is also available in OpenCV. Therefore, the authors should explain the differences between the proposed method and the existing methods in OpenCV.
- In experiments, the authors should also compare the proposed method with OpenCV.
Author Response

(The authors gave the same response as above.)

Round 2
Reviewer 1 Report
table 3
2.1 <- 2.10
Eq.7 sin( and cos( - large brackets required
Reviewer 3 Report
- Figure 4 should be revised. In Fig. 4, the patrol phase is not mentioned in the paper. Moreover, the search phase is missing in Fig. 4.
- At Line 279, the symbols Clength_nm and Clength_nj should be the same as those in Equation (6).
- In Equation (7), arccot(y0n/x0n) should be arccot(yon/xon).
- OpenCV also provides several PnP algorithms, such as AP3P, UPNP, IPPE, etc. The authors should also compare their method with these PnP methods in their experiments.
